



# Analyzing the PMIP4-CMIP6 ensemble: a workflow and tool (pmip_p2fvar_analyzer v1)

Anni Zhao[1], Chris M. Brierley[1], Zhiyi Jiang[1], Rachel Eyles[1], Damián Oyarzún[1], and Jose Gomez-Dans[1]

[1]Dept of Geography, University College London, London, WC1E 6BT, UK

**Correspondence:** Anni Zhao (anni.zhao.16@ucl.ac.uk)

**Abstract.** Experiment outputs are now available from the Coupled Model Intercomparison Project's $6^{th}$ phase (CMIP6) and the past climate experiments defined in the Model Intercomparison Project's $4^{th}$ phase (PMIP4). All of this output is freely available from the Earth System Grid Federation (ESGF). Yet there are overheads in analysing this resource that may prove complicated or prohibitive. Here we document the steps taken by ourselves to produce ensemble analyses covering past and

future simulations. We outline the strategy used to curate, adjust the monthly calendar aggregation and process the information downloaded from the ESGF. The results of these steps were used to perform analysis for several of the initial publications arising from PMIP4. We provide post-processed fields for each simulation, such as climatologies and common measures of variability. Example scripts used to visualise and analyse these fields is provided for several important case studies.

## 1 Introduction

Paleoclimate modelling has long been used to understand the mechanisms of past climate changes, and also has served as a tool to test the out-of-sample boundary conditions and forcings like high atmospheric $CO_2$ concentration that are used in future climate change projections (e.g. Harrison et al., 2014, 2015; Schmidt et al., 2014). The Paleoclimate Model Intercomparison Project, now in its fourth phase (PMIP4; Kageyama et al., 2018) is a project endorsed by the Coupled Model Intercomparison Project phase 6 (CMIP6; Eyring et al., 2016), which aims to analyse and understand the differences between model simulations

of past climates. PMIP4 has been updated from its earlier phase PMIP3 (Braconnot et al., 2012) by including additional past warm periods (Fig. 1) and running improved forcings and boundary conditions by the new generation of climate models (Kageyama et al., 2018; Eyring et al., 2016).

The mid-Holocene (6000 years ago) and the Last Interglacial (127,000 years ago) are characterised by altered seasonal and

latitudinal distribution of incoming solar radiation when the Earth's orbits were different from modern. The *midHolocene* and *lig127k* experiments are Tier 1 PMIP4-CMIP6 simulations (Fig. 1); designed to examine the model response to changes in the Earth's orbit in periods when the atmospheric greenhouse gas concentrations were similar to the preindustrial level and the topographies were also similar to modern. Otto-Bliesner et al. (2017) described the protocols and specific information for the two experiments in detail. Brierley et al. (2020) summarised the large features in the PMIP4-CMIP6 *midHolocene* simulations

and the changes since the PMIP3-CMIP5 generation. Features in the PMIP4-CMIP6 *lig127k* ensemble have been analysed by



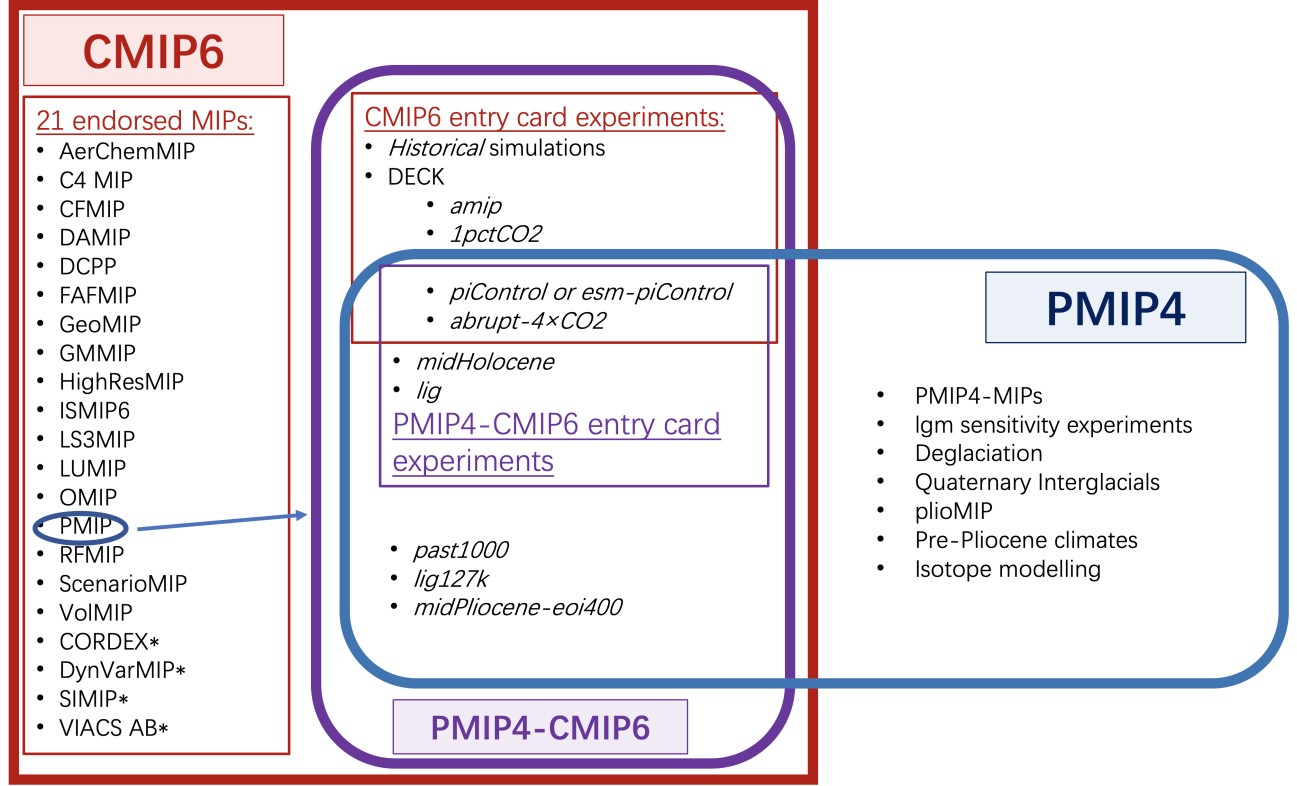

**Figure 1. Schematic diagram illustrating the PMIP4-CMIP6 experiments and their relationship to CMIP6 and PMIP4, according to Eyring et al. (2016) and Kageyama et al. (2018)**

Otto-Bliesner et al. (2021). These two ensembles within PMIP4 have contributed to the text in several chapters (Gulev et al., 2021; Eyring et al., 2021; Douville et al., 2021; Fox-Kemper et al., 2021) in the latest Sixth Assessment Report (AR6) of the Intergovernmental Panel on Climate Change (IPCC), as well as figures (Eyring et al., 2021) .

Here we provide a detailed description of the workflow and scripts that have been used to create figures in the primary description papers of the latest PMIP4-CMIP6 *midHolocene* (Brierley et al., 2020) and *lig127k* (Otto-Bliesner et al., 2021) experiments. The scripts have also been used for figures in past2future multi-experiment papers on the PMIP3-CMIP5 tropical Atlantic inter-annual variability (Brierley and Wainer, 2018), PMIP3-CMIP5/6 climate variability (Rehfeld et al., 2020) and PMIP3-CMIP5 and PMIP4-CMIP6 ENSO (Brown et al., 2020) and contributed to Figures 3.2b and 3.11 in IPCC AR6 WG1

Chapter 3 (Eyring et al., 2021). In section 2, we give a description of the PMIP4 output availability and a discussion of how to access data, along with an evaluation of how and when to apply the `PaleoCalAdjust` software. The following section describes the Climate Variability Diagnostics Package (CVDP) and how it has been modified for PMIP4 purposed. The general





**Table 1.** List of variables in the *piControl, midHolocene* and *lig127k* simulations that were downloaded from the ESGF.

| Long name | Variable name | Unit | Table ID |
|-----------|---------------|------|----------|
| Near-Surface Air Temperature | tas | K | Amon |
| Surface Temperature | ts | K | Amon |
| Sea Level Pressure | psl | Pa | Amon |
| Precipitation | pr | kg m-2 s-1 | Amon |
| Ocean Y Overturning Mass Streamfunction | msftyz | kg s$^{-1}$ | Omon |
| Sea-Ice Area Percentage (Ocean Grid) | siconc | % | SImon |

Information follows
http://proj.badc.rl.ac.uk/svn/exarch/CMIP6dreq/tags/latest/dreqPy/docs/CMIP6_MIP_tables.xlsx

NCL and python routines (as pmip_p2fvar_analyzer v1.0) used in the above papers are described in detail in section 4. Some case studies of possible analyses using the described workflow are given in section 5, followed by a short summary in section

40 6.

## 2 Curating and Collating PMIP4 output

Each model participating in the CMIP6 has uploaded (or is going to) upload their DECK and historical simulations (and endorsed MIPs simulations if available) onto the Earth System Grid Federation (ESGF; Balaji et al., 2018, available at https://esgf-node.llnl.gov/search/cmip6/) in the standard format as required by the CMIP6 Data Request (Juckes et al., 2020). All CMIP6

outputs have been written to netCDF files with one variable stored per file. Full lists of variables in the ESGF controlled vocabulary are available at http://proj.badc.rl.ac.uk/svn/exarch/CMIP6dreq/tags/latest/dreqPy/docs/CMIP6_MIP_tables.xlsx. Users can restrict searching results by selecting appropriate search constraints (e.g. Variable, Experiment ID and Frequency). Table 1 lists the variables and their relevant information that we downloaded from the ESGF and used for analysis. For each PMIP model on the ESGF, data was acquired for every single experiment in the DECK and PMIP4 (see Fig. 1). Only a single variant

was selected for each experiment. Only the FGOALS-g3 *midHolocen*e has multiple runs, and r1i1p1f1 has been selected. There are 4 different forcings available for the IPSL-CM6A-LR (Braconnot et al., 2019) : only the r1i1p1f1 variant has been selected as that relates to the Tier 1 midHolocene protocol.

Since each experiment contained a single ensemble member, a revised database structure was adopted to harmonise both CMIP6 and CMIP5 ensemble conventions. The resulting database only has two directory levels: the top level is taken as the

model, with a sub-directory for each experiment that contains all the outputs listed in Tab. 1. Additionally an 'areacello' fixed variable is stored in each sub-directory for the computation of sea ice area on rotated grids. These were not always deposited on the ESGF for an experiment and needed to be sourced from elsewhere. This curation approach has the added advantage of permitting manual treatment of individual issues. Symbolic links were used to populate this curated ESGF replica where



possible (to avoid the duplication of data files). When many little files were stored on the ESGF, these were concatenated in a

single larger file (using `ncrcat`) to avoid I/O bottlenecks. If output was available for varying years with a simulation, only the common years were used. This curation approach had the additional advantage permitting the inclusion of simulation prior to publication on ESGF, and allowed for a coherent treatment calendar-adjusted files. The file names of the resulting curated directory can be seen at https://github.com/pmip4/UCL_curated_ESGF_replica.

The eccentricity, obliquity and precession at the mid-Holocene and the Last Interglacial were different from those at 1850

CE. Therefore aggregating up daily output to monthly averages using a 'fixed-length' calendar to define the number of days in each month is not appropriate across all the experiments: a 'fixed-angular' calendar should be used instead. Bartlein and Shafer (2019) provide a software, `PaleoCalAdjust`, to convert between the two calendars for simulation output that are produced and stored in general CMIP format. The process taken by the PaleoCalAdjust software is interpolate from non-adjusted monthly averages down to pseudo-daily values and then aggregate these values back up to a 'monthly' resolution for each 30° segments

of the Earth's orbit (see `adjusted_month_lengths.xlsx` on Github for adjusted month lengths). Bartlein and Shafer (2019) evaluated the software's performance for monthly temperature and precipitation variations and showed that in some situations the aliases due to the calendar definition can be larger than the climate change signal. Therefore, Brierley et al. (2020) and Otto-Bliesner et al. (2021) decided to apply the the calendar adjustment when analysing seasonal temperature and precipitation. Regional mean temperature of the warmest month (MTWA) and of the coldest month (MTCO) for the mid-

Holocene in Fig 3.44 in IPCC AR6 WG1 (Eyring et al., 2021) used adjusted monthly temperatures.

Brierley et al. (2020) explored the potential interpolation errors from PaleoCalAdjust on precipitation in monsoon regions (Pollard and Reusch, 2002) by analysing the averaged rain rate during the monsoon season over the South American monsoon domain in the IPSL-CM6A-LR *midHolocene*. As a result, Brierley et al. (2020) decided to not apply the calendar adjustment when analysing monsoon variables despite presenting DJF and JJA precipitation changes that did use it. In general, whether it

is better to use PaleoCalAdjust depends on the steps in the subsequent processing and analysis. If scripts average many months without weighting by month length then we feel it is undesirable to use PaleoCalAdjust. This experience is summarised in the Fig. 2, which provides a recommendation of whether to deploy PaleoCalAdjust dependent on the proposed analyses. The monthly palaeoclimate plots in the IPCC Interactive Atlas (Guiterrez et al., 2021; Gutiérrez et al., 2021) are calendar adjusted, whilst the annual mean fields are computed as a unweighted average of the 12 un-adjusted monthly climatology (Brierley and

Zhao, 2021a). This has the disadvantage that users will not be able recreate the annual mean themselves by downloading and averaging the 12 monthly fields, unless they weight them by the number of days in the month. Gutiérrez et al. (2021) also provide equivalent climatologies for the *midPliocene-eoi400* and *lig127k* ensembles. Subsetted files can be downloaded via the Interactive Atlas, although all the underlying files are also available (Brierley and Zhao, 2021a). It is worth noting that the ensembles available through these sources is only the subset of models who had uploaded output onto the ESGF. This

difference is most marked for the *midPliocene-eoi400* and *lgm*, where images within the IPCC report (Gulev et al., 2021; Eyring et al., 2021) use the larger ensemble of Haywood et al. (2020) and ) that includes non-CMIP6 models. There may also be noticeable differences between PMIP4 *lgm* analysis contained within the report and the Atlas, because of the non-CMIP6 models included by Kageyama et al. (2021). All the analysis of the PMIP simulations discussed in the IPCC Report and its atlas





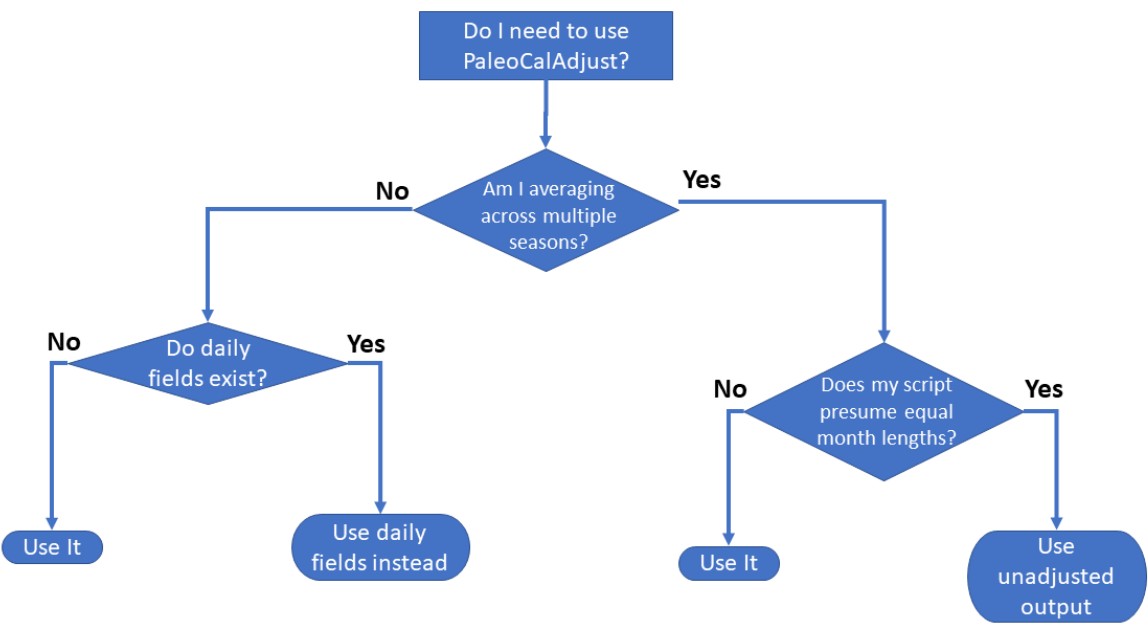

**Figure 2. Flowchart summarising the processes used to determine whether it is desirable to deploy the PaleoCalAdjust software (developed by Bartlein and Shafer, 2019)**

derive ensemble means by considering the models to have an equal weight. This is not the case for future projections in the
report (e.g. Arias et al., 2021) – work is ongoing to determine appropriate weighting for palaeoclimate simulations (Schmidt
et al., 2014) and the present workflow does not yet provide such functionality.

## 3    Post-processing of PMIP4 output

The Climate Variability Diagnostics Package (CVDP; Phillips et al., 2014) was developed by the National Center for Atmo-
spheric Research (NCAR)'s Climate Analysis Section to improve and facilitate the evaluation of major modes of international
climate variability, like El Niño-Southern Oscillation (ENSO) and Atlantic Multi-decadal Oscillation (AMO), in models and
observations. This package computes time spatial patterns, standard deviation and trend maps, climatological fields, power
spectra and time series of climate variabilities in any user-specified model simulations whose files fit CMIP5/CMIP6 out-
put requirements. Analysis results of each model are presented via webpages including a summary table that contains model
performance for key metrics of international variability shown as pattern correlations and RMSE compared to the chosen ob-
servations. Outputs are saved in a netCDF file for later use, which contains the data fields that plotted in each png image. This





package has been used by Fasullo et al. (2020) to evaluate the representation of variability in the CMIP6. The resulting data along with the CVDP source code can be downloaded from http://www2.cesm.ucar.edu/working-groups/cvcwg/cvdp.

CVDP has been adapted for palaeoclimate purposes. Brierley and Wainer (2018) introduced additional coupled modes of variability in the Tropical Atlantic. Brown et al. (2020) altered the compositing of ENSO events to look throughtout the year, not
just DJF. Additionally, they computed La Niña and El Niño composites separately, although these were used in the publication. Otto-Bliesner et al. (2021) introduced the computation of time series of sea ice area, in addition to sea ice extent. Brierley et al. (2020) introduced the calculation of a global monsoon domain following the definition of Wang and Ding (2008); Wang et al. (2011, 2014) by applying two criteria as (a) annual range (local summer - local winter) of precipitation rate is greater than 2 mm d$^{-1}$ and (b) the local summer rainfall exceeds 55% of the annual total. The domain's extent is calculated separately for
each individual monsoon season and its extent and area-averaged rain rate computed. This differs slightly from analyses of future projections, where the domain can be fixed at its present-day extent (Christensen et al., 2013). For regional monsoons, we follow the definition of regional monsoon in IPCC AR5 (Christensen et al., 2013) to compute the monsoon summer rain rate and areal extent for each of the seven regional monsoon. The monsoon diagnostics were also analysed by Otto-Bliesner et al. (2021).

In IPCC AR6, global monsoons in paleoclimate is presented in Chapter 3 Eyring et al. (2021), in which Fig 3.17 shows the global monsoon domain and intensity that is calculated as the ratio of (summer - winter) / (annual mean) for historical simulations. Notably, our criteria are different to the definitions used in the IPCC AR6 WG1 in which the global monsoon is defined as where the annual range of precipitation exceeds 2.5 mm/day as in Kitoh et al. (2013); IPCC (2021). Changes in regional monsoons are included in Chapter 8 (Douville et al., 2021) and Box TS13 Fig 1 (Arias et al., 2021) in IPCC AR6
WG1. However, regional monsoons in AR6 WG1 have been modified from AR5 WG1 which now are defined as six regional monsoons and two domains based on published literature and expert judgement (IPCC, 2021) rather than the seven regional land monsoons in the AR5 WG1 (Christensen et al., 2013) (see IPCC (2021) for details).

There are also some further modifications to the PMIP version of the CVDP package that have not been previously documented.

– The mean and standard deviation spatial fields are only computed over the years used to compute the climatology. This makes no difference unless a `custom_climo` is set (the default is to use all available years). It allows the climate state at a particular point in a transient simulation to be isolated, and is generally only used to select the end of the *abrupt4xco2* and *1pctco2* experiments, or to select the satellite period in the *historical* experiment.

– The temperatures of the warmest month and the coldest month are computed, along with their (interannual) standard
deviations. These are variables that are compiled from pollen-derived reconstructions (Bartlein et al., 2011). As the *mid-Holocene* comparison only features in the supplemental of Brierley et al. (2020), this modification was not documented in the manuscripts methodology.

– The principal-component based definition of the AMOC advocated by Danabasoglu et al. (2012) has been abandoned. We revert to a more-conventional definition: taking the AMOC as the maximum zonal mean streamfunction at 30°N





below 500m. This was originally motivated by the conflation of mean state changes and variability in the *past1000* simulations in the principal-component based definition. Whilst the maximum at 30°N can suffer similar issues, it is simpler to explain and diagnose them.

– Time series of area-average precipitation and surface air temperature are calculated for 58 regions. These consist of 30° latitude bands (over land and sea) as well as the climate regions presented in the IPCC report (AR5 regions; Collins
et al., 2013). These time series are stored as variables named `ipcc_TLA_pr` and `ipcc_TLA_tas` respectively, where TLA is the conventional three letter acronym for the AR5 regions and a text string for the latitude bands. It is worth mentioning that currently the use of the present-day land-sea mask is hardwired into CVDP.

– Monthly time series plots of the new monsoon and area-average diagnostics are written so that only a low resolution mean and spread are visualised for records greater than 150 years. This is more computationally efficient

– The associated web pages have been altered to accommodate the new diagnostics

## 4 Plotting post-processed outputs

A series of scripts have been developed that use the CVDP summary output files as inputs for ensemble analysis. Initial development of these scripts started was in NCL by Brierley and Wainer (2018) to complement the original package. Similar to CVDP , they consist of a library of common functions and then individual NCL scripts that convert these functions into (sets
of) figures. In the build up to CMIP6, the decision was made to pivot this component of the workflow into python. This allows for greater interactivity by users through notebooks and JupyterHubs, as well as greater explanation of any example cases. The repository containing these scripts are given a name of the pmip_p2fvar_analyzer v1 (see the "Code and data availability" section).

### 4.1 NCL

All the NCL scripts require the loading of a series of functions from `cvdp_data.functions.ncl`. These functions (Tab. 2) are themselves divided into three classes: those that return graphics, those that return statistics or tables, and those that are related to the identification and loading of simulation files. Combined these functions are intended to operate with an input of a directory containing all the CVDP summary files and output figures(s) or table(s) directly. All regridding and ensemble averaging is performed on the fly. Functions have been written retrospectively to output the data of these interim steps, but this
is performed through an additional processing of the input directory. The plotting routines allow 'resources' to passed to them, providing a high-level of control over the resulting images. These routines can also accept supra-resources, which are logical flags turning on additional functions, for example CONSISTENCY=True will additionally overlay hatching to indicate when the ensemble is consistent in its signal (as seen in Rehfeld et al., 2020). The NCL routines were used to make all the figures in Brierley and Wainer (2018) and Brown et al. (2020), as well as contributing the first 3 figures of Rehfeld et al. (2020).



**Table 2.** The NCL functions and procedures created to support analyses of the CVDP summary files. They are categorised according to whether they return a graphic, a statistic, or produce no output at all to the user and are intended to be called internally. **Bold** font indicates a procedure, the rest are functions.

| graphics | statistics | internal |
|---|---|---|
| **plotCVDPcomparisonMaps** | extract_latlon_areastat | find_files_wVar |
| **plotDiffEnsMnMaps** | stat_ts_var | read_latlon_var |
| **plotEnsTimeseries** | **createTableGCMsExptsYears†** | read_diff_latlon_var |
| **plot_output_CVDPmap**\* | | find_pair_files_wVar |
| **plot_output_DiffEnsMn**\* | | read_ts |
| | | read_ts_all |
| | | createGCMsNameDictionary |
| | | translateGCMsNameDictionary |

\*returns a netCDF file, as well as an graphic. †returns a LaTeX formatted table.

## 4.2 Python

Similar to the NCL scripts, most of the analysing and plotting processes in python have been written into functions, but they are stored in different scripts according to their purposes respectively instead of being written in a single script. Each script was written in notebook format (see Sect. 4.3 for reasons) and named according to its purpose and usage, with detailed documentations available in the script. These scripts all start with a set of five functions to collect the names of available models that have the variable in the experiment and their corresponding directory and filenames respectively and return a dictionary storing these information as "'model_name':'directory/filename'" (hereafter referred as "target-filename dictionary"), in which function `identify_ensemble_members` requires running `find_experiment_ensemble_members.bash` in `bin` directory to identify available ensemble members (i.e. model names) whose simulations have the target variable in the experiment. Same as the NCL scripts, all regridding and ensemble averaging in python scripts is performed on the fly, and the output can be saved as either a netCDF file or a csv file based on its type, if the user requires. We also have developed a set of plotting schemes in which colours are chosen according to the colour guidelines provided by IPCC-WG1 (available at https://github.com/IPCC-WG1/colormaps; last assess on 17th May 2021). Some examples of possible usage are given in section 5.

## 4.3 Interactive Application

One advantage of the python scripts over the NCL equivalent is their wider user base. The choice of notebooks allows documentation and outputs to be stored with the scripts. A logical next step is to permit users to interact with the scripts. Attempts using Binder to create a cloud computing deployment were found to be underpowered - in part because of the two different





coding languages and any data that would want to be stored along with them. Our solution to this problem is to instead use Docker to create containerised application.

The Docker image contains a platform agnostic set up that can allow users to both create bespoke NCL analyses via a terminal, and act as a Jupyter notebook server to allow uses to fully interact with the notebooks. The docker image is run as a minimal Linux machine inside a container on either a laptop or a cluster. It is not parallelized, so cannot run across multiple cores. However the provision of post-processed summary data means that this is not a problem, although it obviously does rule out the most sophisticated analyses. The Docker image is available at https://github.com/pmip4/pmip_p2fvar_analyzer -

instructions explaining how to use it are included in the documentation at https://pmip-p2fvar-analyzer.readthedocs.io/.

    The repository, and therefore the interactive image, contains a series of summary data of the whole ensemble, as well as a script to download the CVDP output for each individual simulation. These are a series of comma separated value tables (`data_frames` in python terminology), which collect together a single piece of information across each simulation. For example, the file `ESGF_doi.csv` contains the digital object identifier for each simulation – an important piece of information

that should be included in PMIP4 publications to both improve traceability and provide due credit to those having performed the runs. We provide the long-term mean and standard deviation of the interannual timeseries of area averaged temperature and precipitation for each region identified by (Collins et al., 2013), in the AR5_Regions sub-directory. Statistics of the newly-incorporated regional monsoon domains (described in Sect. 3, and plotted in Fig. 7) are included in the `monsoon_domains` sub-directory. Some oft-computed metrics are tabulated in the `common_measures` sub-directory – the global mean surface

temperature also includes each model's CMIP generation and effective climate sensitivity (ECS). The temperature changes averaged for 30° latitude bands over the land and ocean are included in the `tempchange_latbands` sub-directory, along with similar tables taken from the supplementary information of PMIP publications. These files form the basis of Fig 3.2b in IPCC AR6 WG1 (Eyring et al., 2021), and provide the global mean surface temperature in palaeo periods as estimated from climate models in the Technical Summary (Arias et al., 2021) and Chapter 2 (Gulev et al., 2021) in the report. The CVDP

computes many different modes of variability for each simulation (Sect. 3), and the amplitudes of these are tabulated in the `climate_modes` sub-directory. For indices based on area-averaged SST, these are simply the standard deviation of the time series. For modes identified using principal component analysis, the approach of Rehfeld et al. (2020) is adopted and the amplitude is spatial standard deviation of EOF pattern averaged over same region as the analysis.

## 5   Example Uses

By choosing and applying appropriate functions in the scripts, the workflow can produce analyses of temperature, precipitation and monsoon (and other climatic patterns if required). These analyses can involve just the ensemble mean or individual models and return the outputs as figures (spatial maps or scatter plots etc) or data files (as netCDF or csv) as required. Detailed documentation of the scripts and functions are available elsewhere (such as in the accompanying notebooks), so will not be repeated here. Instead we provide some worked examples of outputs to help readers get an idea of the possible options. This

examples have generally already featured in PMIP publications or are alluded to in them.



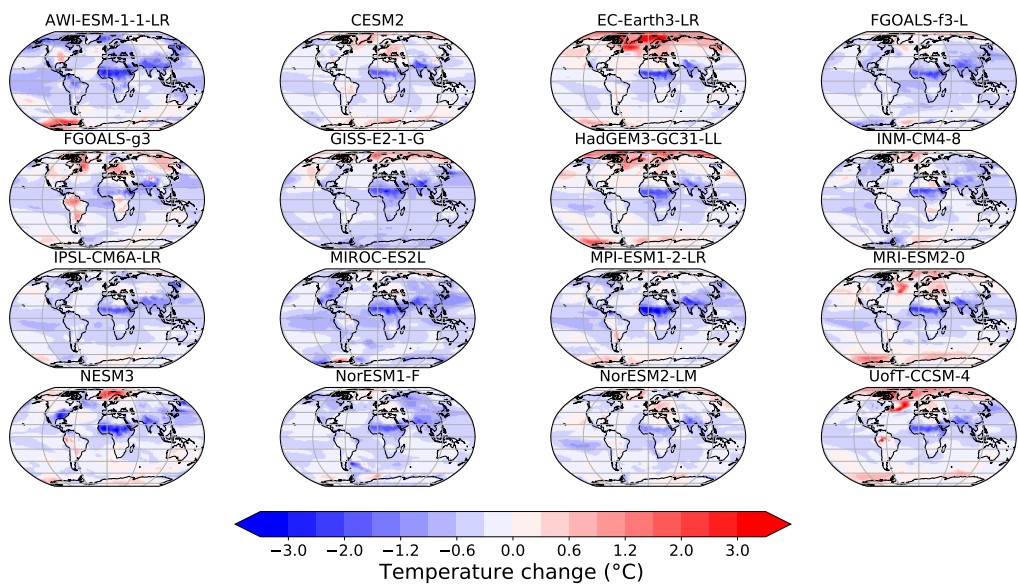

**Figure 3.** Annual mean surface temperature change (°C) in the *midHolocene* as simulated by individual models that features Fig. 4 and Fig.1a in Brierley et al. (2020). Anomalies have been regridded to 1° by 1° resolution.

## 5.1 Plotting spatial patterns

The first example (Fig.3) is the annual mean temperature change in the midHolocene simulations, which shows the patterns of temperature change produced by each model in the PMIP4-CMIP6 *midHolocene* ensemble that features in Brierley et al. (2020). Fig. 3 is generated by using function `Calculation_ensemble_change('PMIP4','midHolocene',`

`'tas_spatialmean_ann','model')` to return a directory containing the regridded annual mean temperature change produced by each model (referred as tas_data) and function `plot_ensemble_tas(tas_data,'ann')` to plot those temperature changes.

The process used to create has four major steps:

1. generate target-filename dictionaries for the *midHolocene* and the *piControl* that have the specified variable;

2. search to see which models occur in both directories, as well as the prescribed list of PMIP4 model names;

3. loop through the matching model's *midHolocene* and *piControl* simulations individually and then compute the difference (i.e. *midHolocene - piControl*);

4. the anomaly is regridded to 1° by 1° resolution (in order to easily calculating the ensemble mean) and then stored in the output dictionary named as the model name.



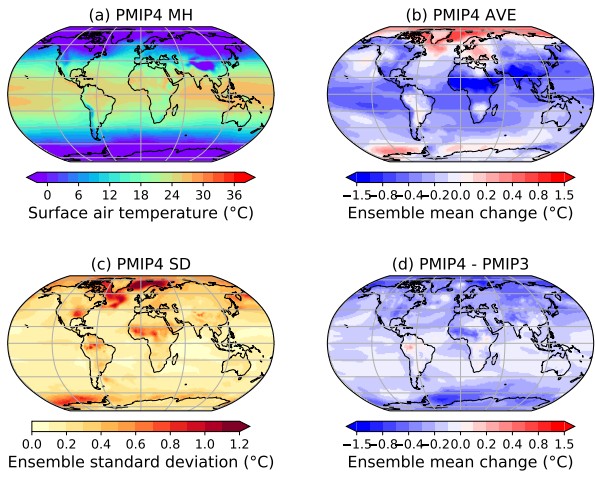

**Figure 4.** Annual mean surface temperature in the *midHolocene* simulations (°C). (a) The ensemble mean, mid-Holocene annual mean surface temperature simulated by PMIP4-CMIP6 *midHolocene* simulations. (b) The ensemble mean, annual mean temeprature changes in PMIP4- CMIP6 (*midHolocene – piControl*) and (c) the inter-model spread, defined as the across-ensemble standard deviation. (e) The difference in the ensemble mean, annual meam temperature changes between the PMIP4-CMIP6 and the PMIP3-CMIP5. Panels (a), (b) and (c) are replotted from panels a, b and e in Fig.1 of Brierley et al. (2020) respectively.

The same function can be used to produce ensemble mean analysis, e.g. the ensemble mean of annual mean surface temperature in the PMIP4-CMIP6 *midHolocene* ensemble shown in Fig. 4, by entering 'mean' instead of 'model' used in example 1. This choice requires an additional step: calculate and return the average and the standard deviation across the ensemble. The NCL programmes do not have the ability to set a flag to determine whether to compute the ensemble mean or individual panels for each ensemble member. Instead they determine the ensemble behaviour depending on the dimension
of files containing the requested input variables. If a difference plot (called using plotDiffEnsMnMaps) detects multiple files for each input, then it will compute an ensemble average of the anomalies. Should the second named experiment resolve to only a single file, then the anomaly between each ensemble member and this file will be computed (this functionality was build to compute the ensemble mean of the biases from a single observational dataset). We provide an example code (multi-panel_plot.ncl) that was used to create Fig. 7 of Brown et al. (2020). This script also demonstrates two of
the supraresources: CONSISTENCY=True provides stippling to highlight where at least two-thirds of the ensemble members show the same sign of anomaly as the ensemble mean; whilst OVERLAY_CONTROL=True adds contour lines showing the ensemble mean of the variable in the second experiment named ("piControl" in this case.).





## 5.2 Plotting oceanic patterns – the Atlantic Meridional Overturning Circulation (AMOC)

The output files from CVDP can be used as an input for computing ensemble mean AMOC changes (mid-Holocene - piControl)
in PMIP4 simulations. The variable called 'amoc_mean_ann', which is analysed with the modified version of CVDP for
individual models, is loaded as an Iris cube for both the mid-Holocene and piControl experiments, then all models are regridded
on to a $1^o$ latitude grid with 61 depth levels between 0-6000m. This process is achieved through Iris cube's interpolation
and regridding schemes called `iris.analysis.Linear()`. Since models can have different names for their dimension
coordinates, basic cube mathematics can not be done directly. Therefore, only the regridded data in each model are extracted for
calculating the differences and averages at this stage. The ensemble mean AMOC change is computed by taking the average of
the regridded changes produced by individual models. After that, the model-averaged data are put back into one of the regridded
models in order to use its dimension coordinates for further plotting. The figure is plotted using 'iris.quickplot' which provides
a visualisation for a cube with a title, x and y labels and a colorbar where appropriate.

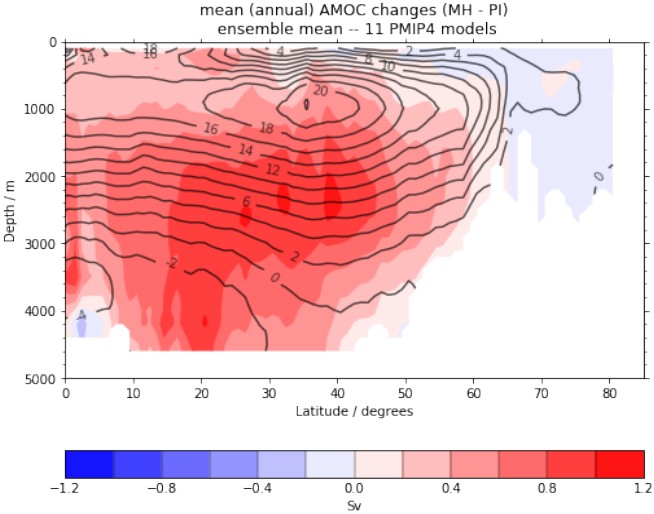

**Figure 5.** Ensemble mean plots for AMOC spatial structure changes between the midHolocene and piControl. 11 PMIP4 models which
performed midHolocene experiment are used for plotting the ensemble means. The overlaid contours (black lines show AMOC strength in
the piControl for locating the maximum AMOC.

In addition, the 'amoc_mean_ann' variable derived from the CVDP output files can also be used as an input for generating
the AMOC profile for a specific latitude, e.g. $30^oN$. This is helpful when comparing the AMOC strength throughout all depths
in different PMIP4 experiments, and was used to create the scatter plot of AMOC values shown in Fig. 10 of Brierley et al.
(2020).



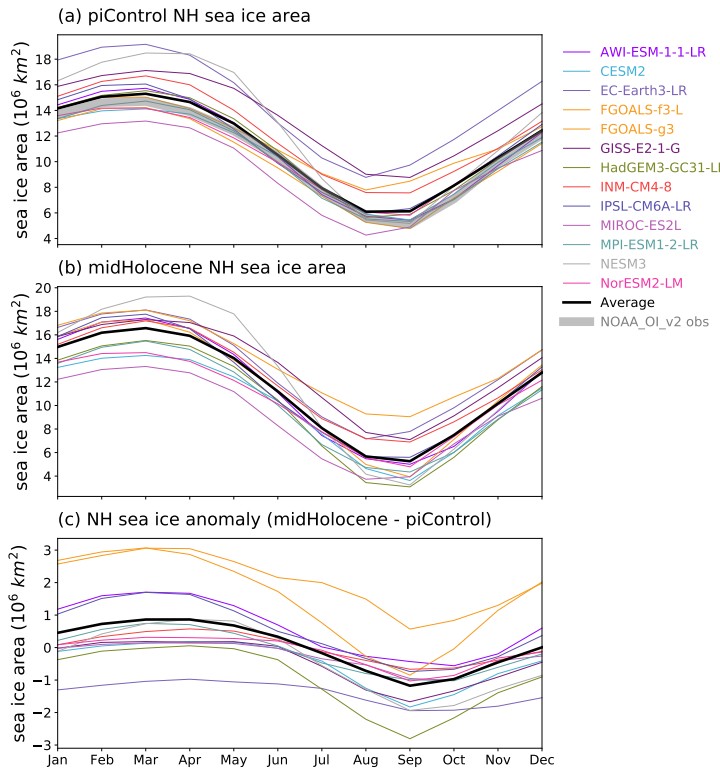

**Figure 6.** Annual cycle of the Arctic sea ice area ($10^6\ km^2$) for the (2) *piControl*, (b) *midHolocene* and (C) their anomaly (*midHolocene - piControl*), replotted from Fig.17 of Otto-Bliesner et al. (2021). The grey contour in panel (a) shows the observed monthly mean sea ice areas from the NOAA_OI_v2 dataset for 1982 - 2001 (Reynolds et al., 2002).

## 5.3 Plotting time series

The CVDP is available to analyse and generate time series data. Fig 6 gives an example of the analysed results of calendar
adjusted monthly mean sea ice areas (defined as sea ice fraction multiplied by the area of each grid cell) in the Arctic in the
*midHolocene* and *piControl* simulations. Lines are coloured following the standard CMIP6 model colour scheme.

## 5.4 Computing statistics

Our scripts can compute statistics from the output files created by the CVDP. The simplest statistics that might be needed at
the length of the data records for a particular variable. The example script `generate_model_table.ncl` interrogates the



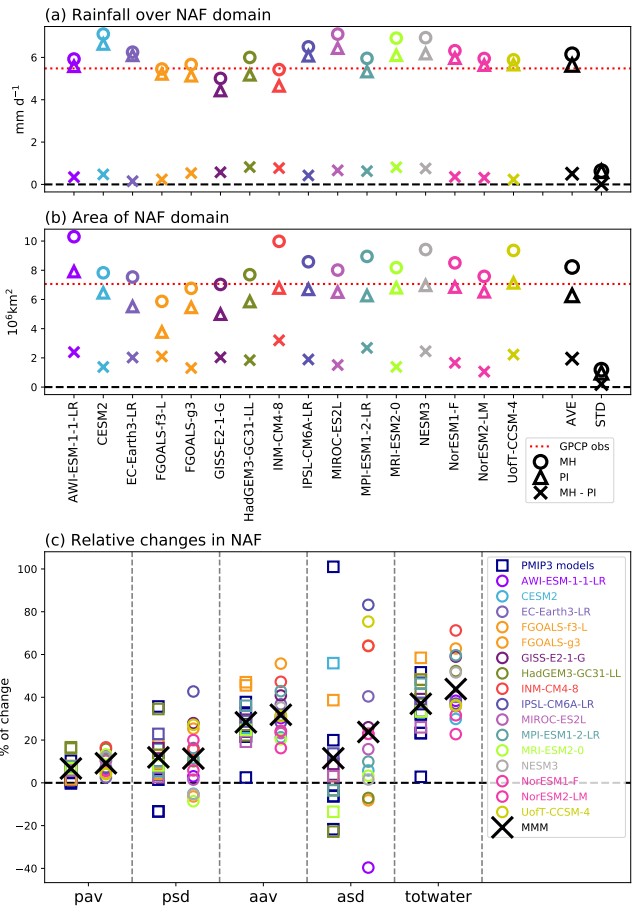

**Figure 7. Changes in the the North African Monsoon (NAF) (a) Area-averaged NAF monsoon summer rain rate (mm d$^{-1}$)** in the *midHolocene* (circles) and *piControl* (triangles) simulations and the difference between them (*midHolocene - piControl* in crosses). **(b) same as panel (a) but for the areal extent of NAF**. The red dotted lines in panels (a) and (b) show the corresponded climatology seen in the 1971-2000 GPCP observational dataset (Adler et al., 2003). **(c)** Relative changes (Brierley et al., 2020) five different monsoon diagnostics beginning from left in order are change in area-averaged monsoon summer rain rate (**pav**), change in the standard deviation of interannual variability in the area-averaged monsoon summer rain rate (**psd**), the change in the areal extent of the NAF monsoon domain (**aav**), change in the standard deviation of interannual variability in the areal extent of the NAF monsoon domain (**asd**) and the percentage change in the total amount of water precipitated in each monsoon season computed as the precipitation rate multiplied by the areal extent (**totwater**). Colours follow the standard colours for CMIP6 models used in the IPCC AR6. If a PMIP3-CMIP5 model (square) is an earlier version of a PMIP4-CMIP6 model, it is coloured same as the PMIP4-CMIP6 one, otherwise coloured by dark blue.



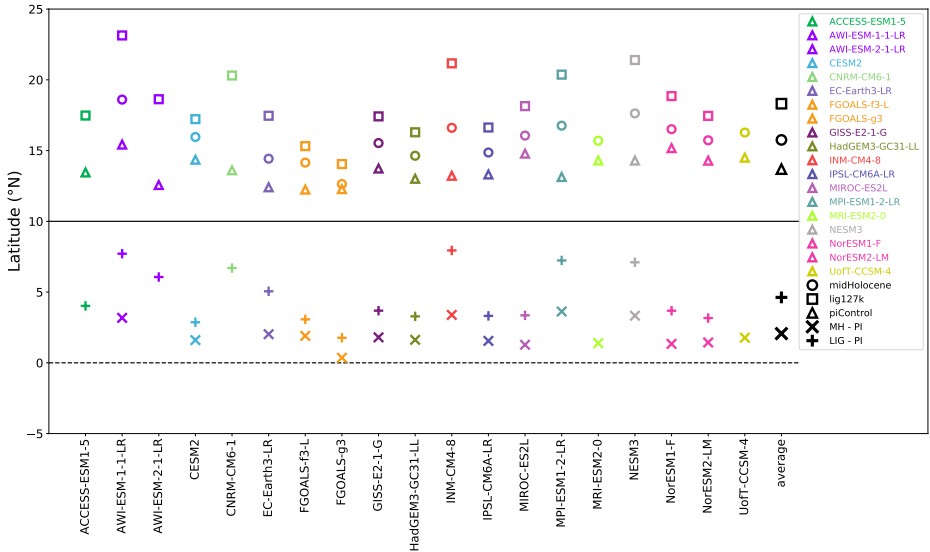

**Figure 8.** NAF expansion in the PMIP4-CMIP6 *midHolocene* and *lig127k* ensembles. The northward monsoon expansion is calculated by determining the change in latitude where the zonal mean summer (MJJAS) rain rate equals to 2 mm d$^{-1}$ over the North Africa (15°W–30°E)

directory to extract the number of years available for every simulation containing a variable called `nino34` performed under series of specified experiments. The resulting table can either be output as a spreadsheet (as comma separated values) or as in a format to be directly inputted into a LaTeX manuscript.

Given a monthly mean precipitation netCDF file as an input, CVDP outputs time series of monsoon rainfall rate (Fig. 7a) and areal extent (Fig. 7b) of each regional land monsoon (Sect. 3). Fig. 7 is the result of this computation for the North African

Monsoon. It uses these time series to compute relative changes in monsoon characteristics in the *midHolocene* experiment and shows the five available diagnostics (Brierley et al., 2020).

It is also possible to compute areal statistics from the latitude-longitude fields. As part of our modifications to the CVDP, area-averaged monthly time series for the AR5 regions are now computed for both precipitation and temperature (Sect. 3). To

cope with the modification of those regions for AR6 (Iturbide et al., 2020), the area-average precipitation changes shown in Fig.3.11 of Eyring et al. (2021) are instead computed from the 2D "pr_spatialmean_ann" variable. An alternate approach to look at the expansion of the North African monsoon is to compute the most northerly latitude to which the monsoon reaches (as in Tab. S2 of Brierley et al., 2020). Fig. 8 gives an example of the NAF expansion in the *midHolocene* and *lig127k* ensembles. We define a series of functions to determine the change in latitude where the zonal mean summer (MJJAS) rain rate (stored



in the files as "monsoon_summer_rainrate") equals to 2 mm d$^{-1}$ over the North Africa (15°W–30°E). See the corresponding notebook for details.

## 6  Summary

The simulations that have been performed for PMIP4-CMIP6, and the large amount of model output available from them, are a great resources for understanding past climates. The procedure by which this model output is analysed as an ensemble can be

time-consuming and involve some methodological decisions. Here we have described the way that our group have chosen to perform our recent analyses (Brierley et al., 2020; Otto-Bliesner et al., 2021; Brierley and Wainer, 2018; Rehfeld et al., 2020; Brown et al., 2020). We document the approach used to obtain and curate PMIP4-CMIP6 simulations, process those outputs via the Climate Variability Diagnostics Package (CVDP), and then continue through to compute ensemble-wide statistics and create ensemble figures. We know from personal experience that replicating the results from published work can often involve

reverse engineering the decisions made by researchers during their data processing, and that this manuscript obviates the need for such effort.

PMIP4 only exists because of the spirit of openness and cooperation within its community, which neatly combines with the IPCC's desire for greater transparency about the figures and data contained within Assessment Report 6. Through documenting our workflow here, we continue in that vein. Hopefully our efforts, such as collation of all the PMIP-related DOIs, make it

easier for others to also be transparent in their research.

The main contribution of this work is not the documentation though - rather it is the provision of post-processed files for each PMIP4-CMIP6 simulation alongside scripts to readily convert them into publication ready figures and tables. The interactive application should increase further lower the barriers to analysis of palaeoclimate model research. We hope that readers are inspired with ideas of potential analyses that they themselves can perform quickly and easily by using the results of our

workflow and scripts as tools to analyse PMIP4-CMIP6 simulations.

*Code and data availability.*  All the codes discussed in the above workflow are available from the PMIP4 organisation on GitHub at https://github.com/pmip4 and from Zenodo at https://doi.org/10.5281/zenodo.5242948 (Brierley and Zhao, 2021b). The original PalCalAdjust software is at https://github.com/pjbartlein/PaleoCalAdjust, whilst the operational version discussed in Sect. 2 can be found at https://github.com/pmip4/PaleoCalAdjust. The original Climate Variability Diagnostics Package software is at https://github.com/NCAR/CVDP-ncl, whilst

the operational version discussed in Sect. 3 can be found at https://github.com/pmip4/CVDP-ncl. All the scripts discussed in Sect. 4.1, Sect. 4.2 and the examples in Sect. 5 can be found within the combined repository at https://github.com/pmip4/pmip_p2fvar_analyzer. These scripts, bundled with relevant data files, can be downloaded as a Docker image, to allow you to interact with them (see the documentation on the repository).

The original climate model output is available from the Earth System Grid Federation. Curated directories can be made available on request

to c.brierley@ucl.ac.uk. A subset of post-processed data is already included in the main repository at https://github.com/pmip4/PMIP_past2future_analyzer, which includes scripts to download the rest if required.



*Author contributions.* A.Z. and C.B. devised and wrote the manuscript. A.Z. wrote all the python scripts, documentation and the majority of the examples presented. C.B. wrote the NCL scripts. Z.Y. contributed the AMOC example. D.O. co-wrote the preliminary codes developed for the 2019 workshop at UCL, where these ideas were first explored. J. G.-D. undertook initial work on the Docker deployment and documentation creation. R.E. was responsible for creating and maintaining the curated ESGF database at UCL during the writing of Brierley et al. (2020) and Otto-Bliesner et al. (2021).

*Competing interests.* The authors declare no competing interests

*Acknowledgements.* This manuscript would not be possible without the generosity of the palaeoclimate modelling community in donating their model output. We hope that they can in turn benefit from some of the code and post-processed data that we describe here. The workflow would not exist without the development of packages by Adam Phillips, Jon Fasullo and Patrick Bartlein. We would like to thank all those in the UCL Geography, especially David Thornalley, for their support and encouragement. C.B. and R.E. were funded in part by NERC (NE/S009736/1) and a UCL Faculty Award.



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
