# Peer review of "Analyzing the PMIP4-CMIP6 collection: a workflow and tool (pmip\_p2fvar\_analyzer v1)"

_Geoscientific Model Development, 2021_

## Author Comment (AC1)

**Curating & Collating**

Download simulation files from the ESGF

Apply calendar adjustment (PaleoCalAdjust v1.0)

Adjusted monthly mean variables

Unadjusted monthly mean variables

**Post-processing**

CVDP -- Compute time spatial patterns, standard deviation and trend maps, climatological fields, power spectra and e times series of climate variabilities.

*CVDP outputs (annual/JJA/DJF tas and pr) are stored in "data_netcdf" directory within the "pmip_p2fvar_analyzer" repository on Github.*

**Computing**

Compute from simulations:
- Bias
- Anomaly
- Inter-annual variability

- *NCL scripts are stored in "ncl_scripts" directory.*
- *Python scripts are stored in "notebooks" directory.*

Calculate ensemble mean bias, change and variation

Compare with reconstructions

**Plotting**

Plot as spatial maps, scatter plots or line plots according to specific requirements.

---

## Author Response (AR1)

**Reply to the reviewers' comments: gmd-2021-290**

A. Zhao *et al.*

**Correspondence:** anni.zhao.16@ucl.ac.uk

**Summary of changes**

We would like to thank the two reviewers for their comments about this manuscript. We have adopted many of the revisions to the figure and text. Both reviewers suggested to replace Figure 2 by a diagram illustrating the processes of this workflow. We have made a flowchart as the new Figure 2 and shall rewrite the relevant paragraphs in text.

5      Blue text below is our response to the editor's comments (reproduced in black).

**Reviewer 1**

This manuscript describes the workflow for the analyses of multi-model ensemble data from the Paleo Model Intercomparison Project (PMIP). It exemplifies the workflow by describing the process of obtaining data, post-process and plot output from the

10   PMIP4 midHolocene and lig127 experiments. It also discusses the specific situation, where paleo data may need an adjustment of the calendar.

The authors discuss their tools and provide a set of scripts and recommendations so that readers can repeat the analyses or apply the toolbox to other experiments. The workflow consists of data-file management, post-processing and analyses, and plotting software. The software is stored in github directories and can be easily accessed. In addition, they provide an application in the

15   "Docker" system.

The manuscript is an interesting contribution. Not often is the workflow from model output to manuscript figures described in a way that other researchers can benefit from. As multi- model analyses become extremely useful with the advent of extensive data repositories like CMIP6, it is a good move to share experience and software to make the process handier. It is also a contribution to make the procedures more transparent.

20   The authors provide an adequate description of the processes involved (but see some critical comments below). The link to the software allows readers to directly try out the methods, for example, by means of Jupyter notebooks.

Overall however, the text should be improved in a revised version. At some places it could be clearer and often it is not really obvious that the information given is necessary or not (see below). Since I have no objection on the general content and methodology, I would rate the reviewer's request as "minor revisions".

25   We would like to thank the reviewer for the thoughtful assessment of our manuscript and the helpful suggestions to improve it.

We are, naturally, happy to make the revisions proposed.

So, I suggest another iteration, where the authors address the following issues: General:

I am not sure if Fig. 2 is needed. It takes quite some place, but could also be put in a few words. I would have expected at that place a schematic on the workflow described in the paper.

You are right. We have replaced the Figure 2 by a flowchart illustrating the workflow.

Minor issues:

Ln 33: I doubt that many readers outside PMIP know what "past2future" means.

We suspect that there may even be people inside PMIP, who don't know. We've now revised this sentence to point out that this is a PMIP working group.

Ln 37: purposes

Done

Ln 49: here it should be piControl, not DECK

The DECK has four experiments, and the *piControl* is one of them. We also downloaded files for the rest DECK experiments, so we decide to remain "DECK" here.

Ln 73: the the

We have deleted the extra 'the'.

Ln 82ff: I wonder if the discussion of the IPCC atlas is needed here in such detail. A short cautionary note would be sufficient.

We have now dramatically reduced the amount of text in the this section. The metadata in the Atlas is presently missing, so we have kept some documentation, but only text relevant to the data provided by this manuscript.

Ln 99 and 104: international == internal

This should have read 'interannual' climate variability. We've corrected the word in both places.

Ln 103-105: it is not clear if you describe the original CVDP or your paleo version

This is a feature of the original CVDP that has not been removed from our paleo version. However, there are no relevant paleo observational datasets to create this table for past climates. We have revised this sentence to hopefully make that clear.

Ln 110: The sentence "Additionally ..." is not clear, what is "although referring to? Or do you mean it was NOT used in the publication?

The latter. We have now added the word 'not' into the sentence. Thank you for spotting this error.

Ln 125: again not clear how relevant this is; are you just discussing the AR6 or your tools?

We were trying to explain the coherence and difference in monsoon definitions used by our script and the IPCC AR6. We recognise that our comparison to AR6 was a bit excessive. We have removed this whole paragraph, and instead provided a much more succinct justification as to why we should remain with AR5 regions for palaeo-contexts.

Ln 134: This sounds a bit odd. Of course, the model data are not derived from pollen. I assume you mean that pollen data are often shown as coldest/warmest months and therefore you need to adapt the model output accordingly.

That assumption is correct. We have rephrased that section to make it clearer what was done.

Section 4: It is a bit confusing where NCL and where python is recommended; can the user chose whatever he/she/they like or are specific steps only possible with one or the other?

We were aiming to let the users choose, but don't state that explicitly. We have now added an explicit recommendation that user try the python scripts. There is greater flexibility within the NCL version, but this would only be of help to expert NCL users who can make full use of that flexibility.

Ln 165: to be passed to them.

Added "be"

Section 4.3: I would start here with a general introduction what systems like Docker do. I assume many readers don't know anything.

Thanks for this suggestion. We agree that many people don't know the Docker. We have added an introduction about it.

Ln 191: users

Done

Ln 264: In figure 6 I recommend to use the same y-scale for figs a) and b)

We have adopted this suggestion and replotted Figure 6 which now has the same y-scales in panels a and b.

Ln 269: "is the length.."?

We've changed "at" to "is".

**Reviewer 2**

General comments:

This manuscript describes a collection of datasets (the "curated ESGF replica"), a modified analysis package (a "paleo version" of CVDP), and a set of scripts written in NCL and Python for additional data analyses and visualization. Although not a model

100    or stand-alone software package, it certainly codifies and advances the analysis of "CMIP-like" model- output data sets. The current approach involves downloading hundreds of files, a lot of fixing up and intermediate analysis, and finally summarization and visualization, with each investigator and group applying their own preferred set of tools, often involving other fix- ups and ad hoc analysis steps. The workflow described here goes a long way toward establishing a "best practices" approach to the analysis of CMIP-type data sets in general, and to PMIP paleoclimatic simulations in particular.

105    One thing that is absent from the beginning of the paper is a discussion of the "why do we need this?" question. A naïve reader might wonder if given a bunch of standardized files (from the ESGF), how hard could it be to produce some figures? Although the files may indeed be standardized, the models have, for example, differing resolutions, the simulations are of differing lengths, and there is considerable data-reduction and transformation that can not be transparently implemented using simple scripts, plus there are many, many variables—hence the utility of the present paper. The place for this discussion might

110    be a new intro paragraph to Section 2 (or at the end of the Section 1) that describes the problem/motivation as in the previous sentence, with the solution being the workflow and tools described here. The sentence beginning on line 293 is a good overall summary, but it's at the end of the paper.

In fact, the workflow and tools could be better illustrated—the components are scattered across several GitHub repositories, and on first reading of the manuscript, I didn't immediately see how one would use the "curated ESGF replica" (or where

115    it was). It took plowing around in repositories to figure that out. I think a figure that illustrates the data assembly, curation, development of second-order data sets (i.e. the CVDP output), and production of summaries and illustrations would be good substitute for Fig. 2, which focuses on a different subject.

We have replaced the Figure 2 by a flowchart illustrating the workflow.

There are lot of technical terms, both climate- and IT-related. Readers might benefit from a few short "in-line" definitions on

120    first use, or URLs to appropriate web pages.

The term "ensemble" is used in several distinct ways here, a) to represent the whole collection of simulations, as in the title; b) to describe all of the individual simulations for a particular experiment (i.e. "lig127k ensembles", line 87); and c) to represent multi-model means, e.g. lines 94, 163. I suggest that the whole collection (i.e. PMIP4-CMIP6, PMIP3-CMIP5, etc.) be referred to as the "collection", all of the simulations for a particular experiment as the "ensemble", and the multi-model means as "multi-

125    model means". There's also a little fuzzy usage of "ensembles" (line 89), where the term could mean "ensemble average" or "ensemble members" ("the subset of models").

We would like to thank the reviewer for the kind comments and are happy to make the revisionsproposed.

The paper does need some mechanical work and smoothing out for readability. Specific comments:

130

line 9: "... to test the out-of-sample response to..." It's not the boundary conditions that are being tested, but instead the response of the models (to boundary conditions different from present).

We've rewritten it to "... to test the response of climate models to the out-of-sample boundary conditions and forcings..."

135 line 15: I think the non-MIP-enabled reader might benefit from a short introduction, maybe a sentence, to the overall notion of using multiple models to simulate the climate while adhering to an identical experimental design, and evaluating those simulations using observations.

A sentence has been added. "Model intercomparison projects have become important in climate research, which run multiple models under the same identical experimental design that helps to synthesise simulated climate change across models."

140

line 16: "... improved forcings and boundary conditions by the new generation of climate models...". There's really two things there, updated experimental designs, and new models.

We've rewritten the sentence to "PMIP4 has been updated from its earlier phase PMIP3 (Braconnot et al., 2012) by including additional past warm periods (Fig. 1), updated experimental designs and involvement of the new generation of climate models

145 (Kageyama et al., 2018; Eyring et al., 2016)."

line 23: Not just topography, but also ice-sheet size.

We've added ice-sheet size in text as "... and the topographies and ice sheet size were also similar to modern."

150 line 32: Non-PMIP readers might wonder about the MIP-numbering scheme (and in particular what PMIP3-CMIP5/6 means, as well as "past2future").

We have dramatically increased the caption of Fig.1 in attempt to make the distinctions clearer. We have edited this sentence to point out that past2future is a PMIP working group.

155 line 41: Reverse the order of "Curating and Collating" to reflect the order that the work is done in. Also, I'm not sure "collating" is the right word; it's main definition includes the ideas of critically comparing or arranging items in order, while I think the work that was done here was perhaps more like "collecting" ESGF data, other data, and the results of the CVDP output.

We've adopted this suggestion. The section is now called "Collecting PMIP4 output"

160

line 45: How many source files are there? (i.e. just the PMIP-CMIP ones, not the whole repository).

We have now added a sentence explaining that 9112 variables are available. The number has been sourced from the ESGF data

holdings webpage at https://pcmdi.llnl.gov/CMIP6/ArchiveStatistics/esgf_data_holdings/PMIP/index.html

165    line 50: You might do an in-line explanation of what "r1i1p1f1" means. Same for "areacello" a few lines down.

We've added a sentence "The variant ID of CMIP metadata is defined in a format of 'r1i1p1f1', where r is realisation, i is initialisation, p is physics, f is forcing and each number is the index for the corresponding configuration." to explain the meaning of "r1i1p1f1". The "areacello" has been explained by "(i.e. grid-cell area for ocean variables)".

170    Line 58: "curated replica". I would reverse the order of discussion to state that you sought to build a replica of part of the ESGF data base (and why), and then how it was populated.

That suggestion does make for a better paragraph. We have moved the penultimate sentence to the top of the paragraph and expanded it.

175    line 60: Explain what ncrcat is.

We have rephrased this to cite the NetCDF Operators tool properly, as well as acknowledging that other software may be appropriate.

line 60: "varying years" Does this mean varying length of simulation? Because individual models use fake years for paleo
180    simulations, there's really little overlap in years. Did you use the last, say, 100 years of each simulation as the common period?

This sentence was meant to related to data being posted onto the ESGF for varying amount of years within a simulation - most notably the FGOALS model in CMIP5. It has been rephrased to "Only years for which all output variables are available were used"

185    line 74: "Calculation of regional mean temperature... used the adjusted monthly temperatures."

"Calculation of" has been added to the beginning of the sentence.

line 77: "midHolocene experiment."

"experiment" has been added.

190

line 79: I would substitute "while" for "despite".

"despite" has been replaced by "while".

line 80: "Interactive Atlas" This is the first mention of this, and probably should have URL.

195    A URL has been added.

line 85: "This has the disadvantage..." It's not really a disadvantage, it's just the way it is. Calculating annual mean temperature as the simple average of monthly mean value, without weighting for month length will always yield a different result than calculating the annual average from, say, daily data (unless the months have equal number of days).

We agree. "This has the disadvantage..." has been deleted.

line 89: "ensembles" or "ensemble averages"

We have deleted the whole sentence.

line 101: "time spatial" What's that?

We've deleted "time".

line 102: "climate variabilities" Variables? Also "user-specified set"

We've changed " variabilities" to "variables", and "user-specified model simulations" to "user-specified set".

line 106: "the variability (of something) in the CMIP6 simulations"?

This has been changed to "the climate variability in the CMIP6 simulations".

line 106: "The resulting data" What resulting data?

This referred to the results of Fasullo et al (2020). We have rephrased this sentence make this fact more obvious.

line 109: Replace "look throughout the year" with "other seasons"?

Done

line 100: "... although these were not used in the publication."? Otherwise, I don't understand.

This was in line 110. We've added the "non" in text.

line 115: "each individual monsoon season" Does this mean each year? Or was it intended to say "each individual monsoon region"?

Yes, it means each year. It's now " each individual monsoon season in each year"

line 116: Reword. (Too many "regional monsoons".)

Done. We've deleted "For regional monsoons" at the beginning of the sentence.

line 130-133: Wouldn't you also want to define a fixed number of years for calculating climatologies? The standard error of the mean is strongly dependent on the number of observations.

We can see that advantage of this, but in practice it up to the user of the CVDP to decide whether to define a fixed number of years. It could be readily achieved by setting the `custom_climo` with negative integers. This possibility was initially created by us and fed upstream into the main CVDP version. A fixed number of years was used by Rehfeld et al. (2020) and Otto-Bliesner et al. (2021), but not by (Brierley et al., 2020). We choose here to include all available years in the climatological for equilibrium simulations.

line 136: What modification, and which manuscripts?
There was only a single manuscript, and we had failed to add the apostrophe. The modification was to the code rather than any definitions. We have now altered 'modification' to 'inclusion'. Additionally, the previous sentence has been rephrased in response to the other reviewer's comments.

line 142: ambiguous "them". The conflation of mean-state changes and variability, or the AMOC variations themselves?
We have removed this whole sentence, and edited the previous one to state the issue with original definition more clearly.

line 149: Finish sentence. More computationally efficient that what?
The sentence has been finished by adding " than calcualting and presenting high resolution data."

line 150: Which associated web pages?
Gosh, we forgot to point out where people can look at all the webpages created when running the CVDP. This has now been corrected.

line 156: I don't understand "greater explanation of example cases".
We've changed "greater" to "better".

line 164: I don't understand "retrospectively".
The scripts were originally written to create only the plots, and then the output scripts written subsequently. The readers don't need to know the historical order of the scripts though. We have rephrased to remove "retrospectively".

line 165: "The plotting routines allow 'resources'...". This is a generic feature of NCL and is not specific to the code described here. For the benefit of non-users of NCL, it might be good to include a couple of sentences that describe the basic architecture of the language.
This has been expanded upon now. We hope that its adopting of official NCL terminology hasn't made it even more confusing.

265     line 174: Break up sentence.

The comma has been deleted.

    line 187: Define "Binder" (or provide a URL).

We have added in a UCL now.

270

    line 189: Define "Docker" (or provide a URL), and "bespoke NCL analyses".

We have added a URL for Docker. We got a bit to flowery with our language and changed "create bespoke" to "perform their own".

275     line 197: "These" Ambiguous. "summary data" or "CVDP output"?

It means summary data."These" has been changed to "The summary data".

    line 198: "... collect together a single piece of information..." Single pieces?

We've replaced "a single piece" by "single pieces".

280

    line 204: "the global-mean surface temperature (something) also includes"

The missing word was 'table', and we have now inserted it.

    line 205: "equilibrium climate sensitivity"?

285 We have changed "effective climate sensitivity" to "equilibrium climate sensitivity". As you correctly infer, these values were calculated from abrupt4xCO2 simulations and should be labelled as such. We had also forgotten the main sources of this data, which are now cited.

    line 205: "temperature changes" Does this mean "long-term mean differences"?

290 We have now added 'long-term' to this sentence. They are the absolute temperature of each individual simulation, but the differences can be readily computed from this table.

    line 209: "... climate models as reported in the Technical Summary..."

"as reported" has been added.

295

    line 216: "... and monsoon (something)..."

It's "... and monsoon characteristics..." now.

line 222: I think "temperature change" is ambiguous. These are long-term mean differences, paleo minus present (the usual convention, but not everybody knows that).

You are probably correct. We have altered the sentence to make this explicit now.

Figure 4: No panel (e) in the figure.

You are right. panel (e) should be (d). We've corrected it in the figure caption.

line 251: Define "Iris cube" (or provide a URL).

We now provide a citation for the Iris software, and have rephrased to remove the word 'cube' as that is not necessary.

line 251: "...all models are regridded onto..." Is this the same kind of regridding as in step 4 on page 10? If so, it probably should have been introduced then; if not, why is a different approach used here (and what was used for step 4)?

We couldn't get the xarray regridding to work properly for this latitude-depth system, so needed to resort to Iris for this. This comment has brought to light our oversight in not mentioning that importance of xarray to the python routines. This has now been acknowledged in section 4.2.

line 253: Why this particular scheme, and not iris.analysis.AreaWeighted()? Interpolation of AMOC data would seem to be an instance where conserving mass or volume would be good.

Although it is often the case that a point-based regridding scheme (such as iris.analysis.Linear() scheme that we use here) is not appropriate when we need to conserve quantities when regridding, and that is the case when the iris.analysis.AreaWeighted() is considered for a conservative regridding scheme. However, the AreaWeighted scheme need the coordinates to be bounded, otherwise we guess the bounds based on the coordinates' point values. Unfortunately the processing by CVDP removes the bounds for all dimensions. This is not a problem for the horizontal grid (longitude and latitude coordinates) from which we can readily guess the bounds. However for the AMOC grid (non-horizontal coordinates) this is not possible due to the non-linear vertical coordinate. Therefore, we choose to use the Linear() scheme which can be applied to any coordinate that satisfying the prerequisites of the chosen scheme for regridding using the Iris Package. We anticipate the errors induced by this choice will be relatively minor.

Figure 6: Should it be "grey line"?

Yes. It has been corrected.

line 266: "the standard CMIP6 model colour scheme" The previous citation to the color schemes only defines the colors, not their assignment to individual models.

https://github.com/IPCC-WG1/colormaps/blob/master/CMIP6_color.xlsx defines the colours for individual models. We've added

a URL to it.

335 Figure 8: I think this figure would "read" better if it were broken into two panels: the top one showing the latitude of the monsoon extent, and the bottom the experiment minus control difference in northward extent. The horizontal line at 10 sort of accomplishes that visually, but what's significant about 10?

Figure 8 has been updated as suggested.

340 line 274: "land monsoon" What does this mean?

The definition of the monsoon regions we've adopted means hat whilst the global monsoon domain exists over both land and sea, the regional domains only exist on land. We have used "land monsoon" to acknowledge this, but it obviously causes more confusion. We've chosen to remove word "land" instead.

345 line 275: "changes in monsoon characteristics relative to present"?

It has been revised as "changes in monsoon characteristics relative to the *piControl*"

line 279: "... area-averaged monthly time series..." Replace with "area-weighted average monthly time series."

Done

350

line 283: "NAF expansion" It's not the region (NAF) that's expanding, but the areal extent of the monsoon in that region.

"expansion" has been replaced by "poleward extension of the areal extent of the NAF".

line 295: "...this manuscript obviates.... The manuscript surely helps explain things, but it's the modification of the CVDP

355 and the NCL and Python scripts that will reduce the need for reverse engineering.

Again this was us trying to use too flowery language. We have rephrased to "...this work removes...".

Technical comments: line 3: "there is overhead"?

We've changed "there are overheads" to "there is overhead".

360

line 21: Delete semicolon.

Done

line 24: "large-scale"

365 Done

line 29: "as well as to figures in that report"

"as well as figures" has been changed to "as well as to figures in that report" as suggested.

370    line 36: "the following section" Specifically give the section number (for parallelism).

The section number has been given.

line 56: "This was"?

Yes. Have corrected it in text.

375

line 65: Delete "up"

Done

line 67: Delete "a"

380    Done

line 68: "stored in the general"

Done

385    line 68: Reword: "The approach taken by the PaleoCalAdjust software is to interpolate from non-adjusted monthly averages to pseudo-daily values, and then to aggregate those values back up to 'monthly' resolution for each 30° segment of Earth's orbit..."

It has been reworded as suggested.

390    line 87: Begin new paragraph at "Subsetted files...". The theme really changes here...

We've deleted this sentence and rewritten the remaining paragraph.

line 92: Reword: "because Kageyama et al. (2021) included non-CMIP models."

It has been reworded as "... and Kageyama et al. (2021) that included non-CMIP6 models."

395

line 99: "interannual" See also line 104.

"international" has been replaced by "interannual" at both places.

line 105: Reword: "For later use, output is saved in netCDF files that contain the data fields that are plotted in each .png

400    image."

The sentence has been reworded as "For later use, output is saved in a netCDF file that contain the data fields that are plotted

in each .png image" as suggested.

line 108: Throughout, CVDP is sometimes used with an article (i.e. "the CVDP"); and sometimes not (e.g. "CVDP", as here). It would be good to standardize.

All CVDP in text has been used with an article.

line 112: Replace semi-colon with a comma.

We've corrected it.

line 113: Delete "as".

Done

line 121: Delete "that is"

Done

line 122: Replace "are different to" with "differs from".

Done

line 123: Replace "is defined as where" with "is defined as the region where".

Done

line 153: Delete "started".

"started" has been deleted.

line 154: Delete "then".

Done

line 155: The Google suggests "Python" should be capitalized.

All 'Python' in text have been capitalised.

line 157: Replace "are given the name" with "named".

We've corrected it.

435      line 160: Delete "the loading of"

Done

line 162: These functions are intended to operate on a directory containing the CMIP summary files..."

Done

440

line 172: Delete "respectively".

Done

line 173: "documentation"

445  We've corrected it.

line 182: "accessed"

We've corrected it.

450  line 185: "Python notebooks allow documentation and output to be stored..."

Done

line 187: "cloud-computing"

We've corrected it.

455

line 188: Replace "would want to" with "should".

We've corrected it.

line 189: "a containerized"

460  "a" has been added before "containerised"

line 190: "platform-agnostic"

We've corrected it.

465  line 191: Replace "uses" with "user".

Done

line 191: "and to act as a Jupyter notebook server that allows.." (for parallelism)

We've revised to sentence for parallelism.

470

line 191: "Docker"

Done

line 197: "comma-separated"

475 We've corrected it.

line 201: "area-averaged"

We've corrected it.

480 line 202: "newly created"

"incorporated" has been replaced by "created".

line 213: "is the spatial"

"the" has been added.

485

line 219: "output"

"s" has been removed.

line 219: "These examples have generally already been featured..."

490 "This" has been replaced by "These".

line 224: "the function"

"the" has been added.

495 line 226: "the function"

Done

line 228: "The process used to create (these images? these directories?)..."

"Fig. 3" has been added.

500

line 229: This list isn't parallel. Change (4) to "regrid the difference". ("Anomaly" in climatology generally refers to a particular observation minus its long-term mean.)

You are right. "the anomaly is regridded..." has been changed to "regrid the difference" as suggested.

505 line 243: "built"

"build" has been changed to "built".

line 243: "We provide example code..."

"an" has been removed.

510

line 249: delete "an"; hyphenate "ensemble-mean"

Done

line 264: "The CVDP can also be used to analyse..."

515 "is available" has been rewritten to " can also be used".

line 268: Replace "at" with "is"?

The 'at' is replaced by 'is' now.

520 Figure 8: "NAF monsoon"

"monsoon" has been added after "NAF".

line 271: "or in a format"

"as" has been deleted.

525

line 285: delete "to"

Done

**References**

Braconnot, P., Harrison, S. P., Kageyama, M., Bartlein, P. J., Masson-Delmotte, V., Abe-Ouchi, A., Otto-Bliesner, B., and Zhao, Y.: Evaluation of climate models using palaeoclimatic data, Nature Climate Change, 2, 417–424, https://doi.org/10.1038/nclimate1456, 2012.

Brierley, C. M., Zhao, A., Harrison, S. P., Braconnot, P., Williams, C. J. R., Thornalley, D. J. R., Shi, X., Peterschmitt, J.-Y., Ohgaito, R., Kaufman, D. S., Kageyama, M., Hargreaves, J. C., Erb, M. P., Emile-Geay, J., D'Agostino, R., Chandan, D., Carré, M., Bartlein, P. J., Zheng, W., Zhang, Z., Zhang, Q., Yang, H., Volodin, E. M., Tomas, R. A., Routson, C., Peltier, W. R., Otto-Bliesner, B., Morozova, P. A., McKay, N. P., Lohmann, G., Legrande, A. N., Guo, C., Cao, J., Brady, E., Annan, J. D., and Abe-Ouchi, A.: Large-scale features and evaluation of the PMIP4-CMIP6 *midHolocene* simulations, Climate of the Past, 16, 1847–1872, https://doi.org/10.5194/cp-16-1847-2020, https://cp.copernicus.org/articles/16/1847/2020/, 2020.

Eyring, V., Bony, S., Meehl, G. A., Senior, C. A., Stevens, B., Stouffer, R. J., and Taylor, K. E.: Overview of the Coupled Model Intercomparison Project Phase 6 (CMIP6) experimental design and organization., Geoscientific Model Development, 9, https://doi.org/10.5194/gmd-9-1937-2016, 2016.

Kageyama, M., Braconnot, P., Harrison, S. P., Haywood, A. M., Jungclaus, J. H., Otto-Bliesner, B. L., Abe-Ouchi, A., Albani, S., Bartlein, P. J., and Brierley, C.: The PMIP4 contribution to CMIP6-Part 1: Overview and over-arching analysis plan, Geoscientific Model Development, 11, 1033–1057, https://doi.org/10.5194/gmd-11-1033-2018, 2018.

Kageyama, M., Harrison, S. P., Kapsch, M.-L., Lofverstrom, M., Lora, J. M., Mikolajewicz, U., Sherriff-Tadano, S., Vadsaria, T., Abe-Ouchi, A., Bouttes, N., Chandan, D., Gregoire, L. J., Ivanovic, R. F., Izumi, K., LeGrande, A. N., Lhardy, F., Lohmann, G., Morozova, P. A., Ohgaito, R., Paul, A., Peltier, W. R., Poulsen, C. J., Quiquet, A., Roche, D. M., Shi, X., Tierney, J. E., Valdes, P. J., Volodin, E., and Zhu, J.: The PMIP4 Last Glacial Maximum experiments: preliminary results and comparison with the PMIP3 simulations, Climate of the Past, 17, 1065–1089, https://doi.org/10.5194/cp-17-1065-2021, https://cp.copernicus.org/articles/17/1065/2021/, 2021.

Otto-Bliesner, B. L., Brady, E. C., Zhao, A., Brierley, C. M., Axford, Y., Capron, E., Govin, A., Hoffman, J. S., Isaacs, E., Kageyama, M., Scussolini, P., Tzedakis, P. C., Williams, C. J. R., Wolff, E., Abe-Ouchi, A., Braconnot, P., Ramos Buarque, S., Cao, J., de Vernal, A., Guarino, M. V., Guo, C., LeGrande, A. N., Lohmann, G., Meissner, K. J., Menviel, L., Morozova, P. A., Nisancioglu, K. H., O'ishi, R., Salas y Mélia, D., Shi, X., Sicard, M., Sime, L., Stepanek, C., Tomas, R., Volodin, E., Yeung, N. K. H., Zhang, Q., Zhang, Z., and Zheng, W.: Large-scale features of Last Interglacial climate: results from evaluating the *lig127k* simulations for the Coupled Model Intercomparison Project (CMIP6)–Paleoclimate Modeling Intercomparison Project (PMIP4), Climate of the Past, 17, 63–94, https://doi.org/10.5194/cp-17-63-2021, https://cp.copernicus.org/articles/17/63/2021/, 2021.

Rehfeld, K., Hébert, R., Lora, J. M., Lofverstrom, M., and Brierley, C. M.: Variability of surface climate in simulations of past and future, Earth System Dynamics, 11, 447–468, https://doi.org/10.5194/esd-11-447-2020, https://esd.copernicus.org/articles/11/447/2020/, 2020.

---

## Editor Decision (ED1)

Editor review of
Analyzing the PMIP4-CMIP6 ensemble: a workflow and tool (pmip_p2fvar_analyzer v1)
by Zhao et al

1) Code and Data availability : I am puzzled by this section. In the first sentence, you write : « All the codes discussed in the above workflow are available ... and from Zenodo at https://doi.org/10.5281/zenodo.5242948 ». But then in the rest of the paragraph, you give links to github repository for different parts of the software used in the paper ; does this mean that the corresponding software is not available on Zenodo ? That should not be so. Please clarify and make sure that all software and codes used in the workflow described is available on Zenodo.

2) Remarks on your responses to the reviewer's comments (the numbering used refers to the number of the line in the document gmd-2021-290-author_response-version2.pdf) :

- L105 : I think you did not address the reviewer's comment on « Why do we need this ». Please do so.
- L127 : I don't see anywhere in the text where you changed "ensemble" for "collection" or "multi-model" as proposed by the reviewer
- L152 : Figure 1 is missing. Reintroduce Figure 1 and explain what was changed in its captions.
- L161 : You added "9112 variables had been uploaded to PMIP4 by Dec 2021" ; I think this does not answer the referee's comment. What he wants to know is how many files/variables should be uploaded per model for PMIP-CMIP.
- L170 : I don't see any sentence moved to the top of the paragraph starting with "Since each experiment …"; please be more precise.
- L175 : I don't see the need to add "in this instance" ; for me it is redundant with "in a single larger file"; please remove it .
- L197 : I don't think the reviewer meant to remove the whole sentence. Please put it back, only changing "This has the disadvantage ..." by "This means ..."
- L225 : Change "in each year" for "for each year".
- L227 : Change "delineation of regional monsoons adopted" for "delineation adopted"
- L250 : In the sentence added, I guess "from" should be "form" ? You write « previously called the 'PMIP Variability Database' » but I don't see where this is done in the manuscript?
- L265 : You did not answer the remark to break up this 5-line long sentence; please do so.
- L271 : Put "https://mybinder.org/" between parentheses and also few lines below put "https://docker.com/" between parentheses.
- L290 : remove the comma after "long-term"
- L300 : as you have "differences" in your sentence, change "minus" for "and"
- L310 : you are discussing the use of xarray but not answering the referee's question about whether or not the regridding is the same kind of regridding as in step 4 on page 10;m please do so
- L330 : I think the mention "last assessed on 30th Nov 2021" is not useful here; consider removing it.
- L335 : What does "The latitude of the boundary is calculated by determining the change in latitude where the zonal mean summer (MJJAS) rain rate equals …" Shouldn't it be simply "The boundary is the latitude where the zonal mean summer (MJJAS) rain rate equals …"

- L351 : Change "to which the monsoon reaches" by "reached by the monsoon"
- L356 : What work? Replace by "the work presented in this manuscript"?
- L359 : You forgot to change "are" by "is"
- L401 : « contain » should be « contains »

3) Remarks directly on your manuscript (the numbering used here below refers to the number of the line in gmd-2021-290-ATC2.pdf) :

- L12 : remove the semi-colon after « (Fig. 1) »
- L17 : I think « PMIP3-PMIP5 » should be replaced by « PMIP3-CMIP5 »
- L89 : replace « who » by « which »
- L119 : I think the possesive form « 's » should be used only for persons, change « domain's extent » for « domain extent »
- L145 : same as above : change « manuscript's methodology » for « manuscript methodology »
- L173 : Put capital P for Python
- L184 : add missing parenthesis after « variable »
- L215 : as you are using « as well as providing » later in the sentence, I think « storing » would read better than « that stores »
- L241 : What does « oft-computed » mean?
- L252 : « Examples uses » instead of « Example Uses »

---

## Author Response (AR2)

**Reply to the editor's comments: gmd-2021-290**

A. Zhao *et al.*

**Correspondence:** anni.zhao.16@ucl.ac.uk

**Summary of changes**

We would like to thank the editor for the remarks about our response to the two reviewers' comments and the remaining errors in our manuscript. We adopted most of the remarks.

Blue text below is our response to the editor's comments (reproduced in black).

Dear Author, I would like to thank you for the revised manuscript that answers most of the point raised by the referees. However, before considering publication of your manuscript, I would like you to consider the following points and produce a new manuscript answering them. With best regards, Sophie Valcke

Thank you, Sophie. Your comments are really helpful. We'd happy to revise our manuscript according to your remarks.

Editor review of Analyzing the PMIP4-CMIP6 ensemble: a workflow and tool (pmip_p2fvar_analyzer v1) by Zhao et al

1) Code and Data availability : I am puzzled by this section. In the first sentence, you write : « All the codes discussed in the above workflow are available ... and from Zenodo at https://doi.org/10.5281/zenodo.5242948 ». But then in the rest of the paragraph, you give links to github repository for different parts of the software used in the paper ; does this mean that the corresponding software is not available on Zenodo ? That should not be so. Please clarify and make sure that all software and codes used in the workflow described is available on Zenodo.

Sorry for causing your confusion. We've created Zendo dois for all the corresponding software and added them after the github repositories.

2) Remarks on your responses to the reviewer's comments (the numbering used refers to the number of the line in the document gmd-2021-290-author_response-version2.pdf):

L105 : I think you did not address the reviewer's comment on « Why do we need this ». Please do so.

A paragraph has been added in Section 1 to introduce why we need the workflow described in this manuscript.

L127 : I don't see anywhere in the text where you changed "ensemble" for "collection" or "multi-model" as proposed by the reviewer

Oh gosh! We are really sorry that we forgot to clarify the usage of "ensemble", "collection" and "multi-model" in text. They have been corrected in text and figures.

L152 : Figure 1 is missing. Reintroduce Figure 1 and explain what was changed in its captions.

My apologies. We didn't notice that Fig 1 is missing through the creation of the marked-up version via latexdiff. It's visible in gmd-2021-290-manuscript-version3.pdf. We've now added Fig 1 back and marked the changes in its caption.

L161 : You added "9112 variables had been uploaded to PMIP4 by Dec 2021" ; I think this does not answer the referee's comment. What he wants to know is how many files/variables should be uploaded per model for PMIP-CMIP.

is the total amount of variables that have been uploaded by the whole community. It is difficult to say how many files should be uploaded per model, because each model supports different number of variables and simulations and doesn't contribute to all PMIP experiments. We have now changed it to the URL of the ESGF CMIP6 PMIP data holdings webpage.

L170 : I don't see any sentence moved to the top of the paragraph starting with "Since each experiment ..."; please be more precise.

We didn't realise that. My apologises! We made a mistake by adding the sentences in a wrong version of the manuscript. They have been added correctly now.

L175 : I don't see the need to add "in this instance" ; for me it is redundant with "in a single larger file"; please remove it .

It has been removed.

L197 : I don't think the reviewer meant to remove the whole sentence. Please put it back, only changing "This has the
disadvantage ..." by "This means ..."

Thanks for pointing out our misunderstanding. The whole sentence has been put back and "This has the disadvantage ..." has been replaced by "This means ..."

L225 : Change "in each year" for "for each year".
Done.

L227 : Change "delineation of regional monsoons adopted" for "delineation adopted"

We've deleted "of regional monsoons".

L250 : In the sentence added, I guess "from" should be "form" ? You write « previously called the 'PMIP Variability Database' » but I don't see where this is done in the manuscript?

Sorry for causing confusion. « previously called the 'PMIP Variability Database' » was meant it was used in another published paper written by one of us, Chris Brierley and his colleague. It now has been corrected and referenced to the literature.

L265 : You did not answer the remark to break up this 5-line long sentence; please do so.
The long sentence has been broken up to two.

L271 : Put "https://mybinder.org/" between parentheses and also few lines below put "https://docker.com/" between parentheses.
Brackets have been added.

L290 : remove the comma after "long-term"
We've removed the comma.

L300 : as you have "differences" in your sentence, change "minus" for "and"
"minus" has been replaced by "and".

L310 : you are discussing the use of xarray but not answering the referee's question about whether or not the regridding is the same kind of regridding as in step 4 on page 10;m please do so
Sorry we didn't answer the question. The regridding steps were different, because they were written by different people. Now the codes have been updated, which now use the same kind of regridding as in step 4.

L330 : I think the mention "last assessed on 30th Nov 2021" is not useful here; consider removing it.
We agree. The assess date has been removed.
L335 : What does "The latitude of the boundary is calculated by determining the change in latitude where the zonal mean summer (MJJAS) rain rate equals ..." Shouldn't it be simply "The boundary is the latitude where the zonal mean summer (MJJAS) rain rate equals ..."
Thanks! We've rewritten this sentence following your suggestion.
L351 : Change "to which the monsoon reaches" by "reached by the monsoon"
Done.

L356 : What work? Replace by "the work presented in this manuscript"?
We've changed "this" to "the" and added "presented in this manuscrip" after "work".

L359 : You forgot to change "are" by "is"

"are" has been changed by "is".

L401 : « contain » should be « contains »

A "s" has been added.

3) Remarks directly on your manuscript (the numbering used here below refers to the number of the line in gmd-2021-290-ATC2.pdf) :

L12 : remove the semi-colon after « (Fig. 1) »

";" has been removed.

L17 : I think « PMIP3-PMIP5 » should be replaced by « PMIP3-CMIP5 »

You are right. We've corrected this typo. Thanks!

L89 : replace « who » by « which »

Done.

L119 : I think the possesive form « 's » should be used only for persons, change « domain's extent » for « domain extent »

It has been removed.

L145 : same as above : change « manuscript's methodology » for « manuscript methodology »

Corrected.

L173 : Put capital P for Python

P has been changed to capital.

L184 : add missing parenthesis after « variable »

")" has been added after variable.

L215 : as you are using « as well as providing » later in the sentence, I think « storing » would read better than « that stores »

"that stores" has been changed to "storing" for parallelism.

L241 : What does « oft-computed » mean?

This niche adverb means that the calculation is often performed by the community. We have now changed it some something more generally understandable.

L252 : « Examples uses » instead of « Example Uses »

The section title has been changed.

---

## Author Response (AR3)

**Reply to the editor's comments: gmd-2021-290**

A. Zhao *et al.*

**Correspondence:** anni.zhao.16@ucl.ac.uk

We would like to thank the editor for accepting our manuscript with minor corrections. We have revised the manuscript as required. Blue text below is our response to the editor's comments (reproduced in black).

Dear Author,

Thank you very much for your revised manuscript that answers all my comments. Please include the following minor correction in the next version that you will submit for publication (lines refer to the version 4 of your manuscript):

Thank you!

L.36: "would be" -> "is" (would be sounds as if you did not provide what is needed)

Thanks for pointing this out. We've changed "would be" to "is".

L.49: "(or is going to) upload" -> "(or is going to upload)"

We have moved the ")" to its right place.

L.164: "python" -> "Python"

Done.

L. 325: "at different repositories" -> "in different repositories"

"at" has been changed to "in".

L. 333: "can be count" -> "can be found"

We have corrected this.

Figure 1: Thanks for reintroducing. I think the font of "PMIP4-CMIP6 entry card experiments" should be the same size than "CMIP6 entry card experiments"

Yes, you are right. We have changed the font size to the same. We have also made a few changes (like the font of "PMIP4-CMIP4" and a typo "lig" -> "lgm") to make this figure look better.

With best regards, Sophie Valcke